# Deep Learning Models for Automatic Upper Airway Segmentation and Minimum Cross-Sectional Area Localisation in Two-Dimensional Images

**DOI:** 10.3390/bioengineering10080915

**Published:** 2023-08-02

**Authors:** Guang Chu, Rongzhao Zhang, Yingqing He, Chun Hown Ng, Min Gu, Yiu Yan Leung, Hong He, Yanqi Yang

**Affiliations:** 1Orthodontics, Division of Paediatric Dentistry and Orthodontics, Faculty of Dentistry, The University of Hong Kong, Hong Kong SAR, China; u3006909@connect.hku.hk (G.C.);; 2Department of Computer Science and Engineering, The Hong Kong University of Science and Technology, Hong Kong SAR, China; 3Division of Oral and Maxillofacial Surgery, Faculty of Dentistry, The University of Hong Kong, Hong Kong SAR, China; 4Department of Orthodontics, The State Key Laboratory Breeding Base of Basic Science of Stomatology (Hubei-MOST), Key Laboratory of Oral Biomedicine Ministry of Education, School & Hospital of Stomatology, Wuhan University, Wuhan 430072, China

**Keywords:** artificial intelligence, cone-beam computed tomography, convolutional neural networks, airway segmentation, CSAmin localisation

## Abstract

Objective: To develop and validate convolutional neural network algorithms for automatic upper airway segmentation and minimum cross-sectional area (CSAmin) localisation in two-dimensional (2D) radiographic airway images. Materials and Methods: Two hundred and one 2D airway images acquired using cone-beam computed tomography (CBCT) scanning were randomly assigned to a test group (*n* = 161) to train artificial intelligence (AI) models and a validation group (*n* = 40) to evaluate the accuracy of AI processing. Four AI models, UNet18, UNet36, DeepLab50 and DeepLab101, were trained to automatically segment the upper airway 2D images in the test group. Precision, recall, Intersection over Union, the dice similarity coefficient and size difference were used to evaluate the performance of the AI-driven segmentation models. The CSAmin height in each image was manually determined using three-dimensional CBCT data. The nonlinear mathematical morphology technique was used to calculate the CSAmin level. Height errors were assessed to evaluate the CSAmin localisation accuracy in the validation group. The time consumed for airway segmentation and CSAmin localisation was compared between manual and AI processing methods. Results: The precision of all four segmentation models exceeded 90.0%. No significant differences were found in the accuracy of any AI models. The consistency of CSAmin localisation in specific segments between manual and AI processing was 0.944. AI processing was much more efficient than manual processing in terms of airway segmentation and CSAmin localisation. Conclusions: We successfully developed and validated a fully automatic AI-driven system for upper airway segmentation and CSAmin localisation using 2D radiographic airway images.

## 1. Introduction

Upper airway obstruction is a major cause of sleep-disordered breathing and obstructive sleep apnoea (OSA); this has increased the importance of airway assessment [1,2]. OSA caused by upper airway obstruction is common in the modern global population and has been associated with increased incidences of hypertension, atrial fibrillation, coronary heart disease and stroke [3,4]. OSA reportedly affects 17% of women and 34% of men in the US and has shown a similar prevalence in other countries; however, approximately 85% of patients remain undiagnosed [3,5].

Because the upper airway is anatomically located just behind the oral cavity, airway diseases are often first detected upon radiographic examination by dentists, especially orthodontists. However, the superimposition of the upper airway upon adjacent structures in 2D images, combined with the lack of quantitative measurement standards, makes the preliminary diagnosis of upper airway obstruction using lateral cephalograms (LCs) quite challenging and sometimes inaccurate, even though LC is usually the ‘first-hand’ record for airway evaluation [6,7].

Three-dimensional (3D) computed tomography (CT) and cone-beam CT (CBCT) are more promising techniques than LCs for upper airway assessment. However, only patients with severe craniomaxillofacial deformities are required to undergo large-scale CT scans that cover the upper airway area, owing to the relatively higher radiation dose than 2D radiography. The risks of exposure to high radiation and high costs render 3D screening tools impractical for detecting potential airway problems [8,9]. Therefore, albeit challenging, obtaining sufficient and accurate upper airway information through 2D images, i.e., ‘first-hand’ image records, is important for the early detection of potential airway problems.

Currently, both 2D and 3D airway image analysis are primarily conducted manually, with the processing performed by professional dental/radiographic experts [10,11,12]. These methods are time-consuming, tedious and expert-dependent, which negatively affects the efficiency and accuracy and can lead to missed diagnoses of the disease; all of the above can have significant impacts on human health [13,14]. Recently, artificial intelligence (AI) and deep learning techniques, which employ computers or machines to imitate human logic and cognition to complete a series of intelligent tasks [15], have been widely applied in medical imaging. AI processing is highly efficient because it can extract dental features and swiftly make decisions based on big data; studies have reported its superiority and efficacy in imaging detection compared with manual processing. In dentistry, AI algorithms based on convolutional neural networks (CNNs) have shown remarkable capabilities in numbering and classifying teeth, detecting root fractures and periodontal bone loss and diagnosing dental caries [16,17,18]. However, until now, the effects of AI techniques on upper airway assessment have not been thoroughly explored. Therefore, we aimed to explore the possibilities of applying AI techniques to automatic upper airway assessment to achieve more accurate and efficient evaluations, thereby providing valuable information for the preliminary diagnosis of upper airway obstruction.

The basic step of upper airway assessment is to detect the airway contour and divide it into three segments: the nasopharynx, retropalatal pharynx and retroglossal pharynx [13,19]. The morphology and features of the different segments serve as anatomical data that indicate airway obstruction, if present [9,14]. Automatic and accurate segmentation of the upper airway is fundamental for the application of AI techniques to airway assessment. Furthermore, the minimum cross-sectional area (CSAmin), which denotes the region with the greatest constriction, is a crucial parameter for airway assessment [20,21,22,23]. But CSAmin localisation requires clinicians to reconstruct the upper airway structure in three dimensions and compare the area values across different planes. In recent AI studies, CNNs were used to successfully infer or reconstruct 3D structures based on 2D data after realising large amounts of 3D features and data [24,25]; this would have been impossible to achieve using manual processing (owing to a lack of data from the third dimension).

Therefore, we hypothesised that the development of AI models based on deep learning algorithms can facilitate automatic upper airway segmentation and CSAmin localisation in 2D radiographic airway images, and AI processing can improve the efficiency greatly.

## 2. Materials and Methods

This study was approved by the Institutional Review Board of the University of Hong Kong (IRB reference no: UW 21-519). The methods were conducted in accordance with approved guidelines and regulations. All CBCT scans were obtained for diagnosis and treatment planning. Written informed consent was obtained from patients, and patients were informed that their clinical images may be used for clinical teaching and paper publishing.

### 2.1. Data Collection

CBCT scans from 201 orthodontic patients (76 men and 125 women) aged over 18 years (43.45 ± 19.28 years) were collected from a consecutive sample of the Division of Paediatric Dentistry and Orthodontics, Faculty of Dentistry, the University of Hong Kong, between 1 September, 2016 and 31 December, 2021. The CBCT scan field view covered the whole skull and upper airway contour, with the superior border above the calvarium and inferior border below the epiglottic base. The regions of interest (ROIs) of training data in this study are three segmented regions (the nasopharynx, retropalatal pharynx and retroglossal pharynx) of upper airway in the annotated segmentation map. The exclusion criteria were as follows: (1) patients who presented severe skeletal deformities, facial asymmetry, cleft lip and palate and craniofacial syndrome, which might affect the upper airway morphology and (2) patients with defective CBCT scans (low resolution, improper head posture or unclear airway structures).

The sample ratio of the training group to the test group was set at 4:1 [16,17,18]; the 201 images were randomly assigned to the training group (n = 161) to train AI models or the test group (n = 40) to evaluate the efficacy of AI processing. Standard 4-fold cross-validation was conducted on the training set for basic hyperparameter search, and the test data were kept blind to the model until the final evaluation. The data collection and AI processing flow are presented in Figure 1.

### 2.2. Consistency Tests

A solo examiner (CHU G), who is a well-trained dentist with sufficient experience in dental imaging detection, conducted all the manual image annotation work including midsagittal plane capture, upper airway segmentation and CSAmin level localisation. To ensure data validity and reliability, two experienced examiners (CHU G and Ng CH) conducted consistency tests. Intra- and interexaminer reliability were assessed for midsagittal plane capture, image segmentation and CSAmin level detection.

To assess intraexaminer reliability, 10 randomly selected images from all sample data of 201 images were analysed by the examiner (CHU G); after 1 month, the assessment was repeated to test reliability and repeatability. Precision (the ratio of correctly predicted pixels between the two measurements) was used as the main metric to evaluate consistency.

To assess interexaminer reliability, 10 randomly selected images from all sample data of 201 images were analysed by two examiners (CHU G and Ng CH) to test the repeatability of the midsagittal plane capture and CSAmin level detection, respectively.

### 2.3. Determination of the Midsagittal Plane and Image Capture

All of the CBCT scans were obtained using the CBCT device ProMax 3D Mid (Planmeca Oy, Helsinki, Finland). The clinical staff of the Oral and Maxillofacial Radiology Unit of the Faculty of Dentistry, the University of Hong Kong, performed CBCT scanning using a reference ear plug and head posture aligner to maintain a natural head position. Raw CBCT data were saved and extracted using proprietary software (ROMEXIS, v4.4.0.R; Planmeca, Helsinki, Finland).

Then, CBCT scans were converted to a DICOM image format and evaluated using Mimics v19.0 (Materialise, Leuven, Belgium). All the scans were then reconstructed with slices of 0.5 mm thickness and 0.4 mm voxel size. Five reference points (posterior nasal spine, posterior point of vomer and first, second and fourth cervical vertebrae) were manually obtained for each scan to determine the midsagittal planes of the CBCT images (Figure 2). Each of these reference points is defined in Table 1. The determined midsagittal plane images were captured and saved in PNG format (32 bits, 910 × 910 pixels).

### 2.4. Upper Airway Segmentation

#### 2.4.1. Manual Segmentation of the Airway and Data Augmentation

A well-trained dentist (CHU G, solo examiner) segmented the upper airway from the 2D images using the image processing tool Microsoft Paint in Windows 10 system. The airway structure in each image was segmented into three parts [13,19]: the nasopharynx, extending from the nasal turbinate level to the hard palate; the retropalatal pharynx, extending from the hard palate to the margin of the soft palate; and the retroglossal pharynx, extending from the soft palate to the epiglottic base. Each segment of the airway is shown using different colours in Figure 3.

All of the images were resized to obtain a unified resolution of 400 × 400 pixels. Manually annotated segmentation maps were converted into label maps, in which all pixel values were integers between 0 and 3; 0 represented the background region, whereas 1–3 represented the three segmented regions (the nasopharynx, retropalatal pharynx and retroglossal pharynx) in the annotated segmentation map. During the training process, data augmentation techniques were used to increase variability in the training dataset by resizing, cropping, flipping and randomly rotating the images.

#### 2.4.2. AI Segmentation Models

The segmentation architectures UNet and DeepLab v3 were selected as the training models. The UNet and DeepLab network architectures were each implemented at two depths: UNet18, UNet36, DeepLab50 and DeepLab101. These four AI models were then trained for the segmentation tasks. All the models were trained using the Adam optimisation algorithm at 200 epochs. The architecture was trained in the Pytorch framework (learning rate, 0.0001; batch size, 8) to achieve the best possible validation loss. All the processing work was conducted on a PC with an Intel i7-8700 CPU, 32GB RAM and a single Nvidia RTX 2080 Ti GPU with 12G VRAM (Jumbo computer supplies, Hong Kong, China). Figure 4 presents the network architecture of the UNet model.

#### 2.4.3. Evaluation Metrics for Airway Segmentation

The following metrics were used to evaluate the performance of the segmentation models: precision, recall, Intersection over Union (IoU), dice similarity coefficient (DSC) and size difference.

Precision indicated how many of all the predicted pixel results had been correctly predicted.
$$Precision=\frac{TP}{TP+FP}$$

Recall measured how many of all the correct pixel results had been correctly predicted.
$$Recall=\frac{TP}{TP+FN}$$

IoU has been commonly used to evaluate image segmentation tasks; it represents the similarity of segmentation results between the predicted image and ground truth.
$$IoU(A,B)=\frac{A\cap B}{A\cup B}=\frac{TP}{TP+FN+FP}$$

DSC was calculated as twice the overlapping area divided by the sum of the areas of *A* and *B*.
$$DSC=\frac{2*(A\cap B)}{A+B}$$

Size difference measured the gap between the predicted image and ground truth in pixels, i.e., the number of ground truth segmentation pixels subtracted from the number of pixels of the predicted segmentation results.

### 2.5. CSAmin Localisation Task

#### 2.5.1. Manual Determination of CSAmin

To determine the exact CSAmin level in the entire upper airway passage, each 3D upper airway model was reconstructed using Mimics v19.0. The inferred CSAmin could be manually detected through the midsagittal view in reconstructed 3D images. The height level was marked in each image to illustrate the specific localisation of the inferred CSAmin and was recorded as *H_M_* (manually determined height of CSAmin). The work of manual determination of CSAmin was conducted by the solo examiner (CHU G) as stated previously.

#### 2.5.2. AI-Driven Determination of CSAmin

Forty images each, after manual and AI-driven segmentation, were acquired as the two test datasets. The nonlinear mathematical morphological operations comprising erosion and dilation were employed as the postprocessing procedure for the segmentation images. CSAmin was automatically determined after applying erosion and dilation to extract the binary image boundaries.

##### Erosion and Dilation

Erosion is generally used to extract the inner boundary, which refers to the set of pixels comprising boundary points and belonging to a part of the original regions. The inner boundary was extracted using erosion processing to obtain a contraction of the original image. An exclusive OR operation was then conducted between the contraction results and target image, thereby realising the difference set extraction. After erosion processing, the undesired noise was efficiently removed, and the main area was smoothed in the predicted segmentation map.

Dilation is generally used to extract the outer boundary, which refers to the set of pixels adjacent to the boundary points outside the region and is a part of the background. The outer boundary was extracted using dilation processing. An exclusive OR operation was then performed between the dilation result and original target image to obtain the difference set.

##### Computation and Prediction of CSAmin

After erosion and dilation processing, 40 manual segmentation maps (GT_1_) and 40 AI-driven segmentation maps (GT_2_) were considered as the two types of ground truth. The sum of the number of the pixels in each row in GT_1_ and GT_2_ was calculated to obtain the width of the upper airway at different levels. The narrowest slice with the minimum number of pixels was selected and subjected to the original coordinate system. The narrowest heights in the GT_1_ and GT_2_ groups were recorded as $H_{1}$ and $H_{2}$, respectively. The height difference (*L*) between the predicted ground truth and manually determined heights was calculated to compare the performance of CSAmin prediction as follows:$$L_{1}=|H_{1}-H_{M}|\;L_{2}=|H_{2}-H_{M}|$$

### 2.6. Time Comparison

To test the efficiency of the AI models, the time consumed for manual work (CHU G) and AI processing was recorded when performing the segmentation and CSAmin localisation tasks in the test set (40 images). The average time spent on each image was compared between the two processing methods.

### 2.7. Statistical Analysis

Precision (the ratio of correctly predicted pixels between two measures) was assessed to evaluate the consistency in manual segmentation. The intra- and interclass correlation coefficients (ICCs) were calculated to evaluate the intra- and inter-rater agreements of the sagittal view coordinates and *H_M_*, respectively.

To evaluate the performance of the AI-driven segmentation, the chi-square test was then used to compare precision, recall, IoU and DSC across the different segments. And the Kruskal–Wallis test was used to compare the size differences.

To evaluate the CSAmin height prediction performance, the kappa test was used to evaluate the consistency of CSAmin determinations across specific segments and compare it between manual and AI processing. The paired *t*-test and Bland–Altman analysis were used to compare *H_M_*, *H*_1_ and *H_M_*, *H*_2_. The two-sample *t*-test was used to compare the height differences *L*_1_ and *L*_2_. *p* < 0.05 indicated statistical significance. All the data were analysed using Statistical Product and Service Solutions (SPSS) v27.0.

## 3. Results

The average precision reliability values of manual segmentation for the 10 paired images were 0.971, 0.967 and 0.969 in the nasopharynx, retropalatal pharynx and retroglossal pharynx, respectively. The ICCs of the midsagittal plane coordinate were 0.973 and 0.936 for the intra- and inter-rater agreements, respectively. The ICC values of *H_M_* determination for the intra- and inter-rater agreements were 0.989 and 0.954, respectively. All the consistency tests indicated a high degree of intra- and interobserver agreement.

### 3.1. Accuracy Analysis for AI-Driven Upper Airway Segmentation

The accuracy results of the different AI models are presented in Table 2 and Table 3. Overall, the precision and recall reached 90.0–90.6% and 88.0–89.2%, respectively. In general, no significant differences in accuracy were noted for any of the four AI models (*p* = 0.476, 0.562, 0.433, 0.552 and 0.283 for precision, recall, IoU, DSC and size difference, respectively). In terms of the segmentation results, a difference was detected in the size difference only when using the DeepLab50 model. As the size difference results of the UNet36 model were not normally distributed after testing using the Shapiro–Wilk test (*p* = 0.037), the Kruskal–Wallis test was used to compare the size differences. The size difference in the nasopharynx was significantly larger than those in the retropalatal pharynx and retroglossal pharynx (*p* = 0.039). Figure 5 shows the airway segmentation results obtained from manual processing and the UNet18 model.

### 3.2. Accuracy Analysis for AI-Driven CSAmin Localisation

The accuracy of CSAmin localisation in specific segments for the manual (GT_1_) and full AI (GT_2_) processing methods is presented in Table 4. The kappa test revealed that the consistency between the two results was 0.944. Figure 6 shows the CSAmin localisation results for the GT_1_ and GT_2_ groups.

The detailed results of AI performance in the two test sets are presented Table 5; no significant differences were noted between *H*_1_ and *H_M_* (*p* = 0.780) or between *H_2_* and *H_M_* (*p* = 0.295). The height difference between the *H_M_* of manual work and *H_1_* of the first test set was 1.95 ± 2.21 mm, whereas that between the *H_M_* of manual work and *H_2_* of the second test set was 2.53 ± 2.21 mm. Furthermore, Bland–Altman analysis was conducted to compare the difference between *H_M_*, *H*_1_ and *H_M_*, *H*_2_, and showed that 95.00% (38/40) and 90.00% (36/40) of the points were within the 95% conformance level, respectively (Figure 7 and Figure 8). The height error *L_1_* across the 40 subjects ranged from 0 to 12.31 mm; the proportion of errors <3 mm was 82.5% (33 out of 40). The height error *L*_2_ ranged from 0 to 9.89 mm; the proportion of errors <3 mm was 70.0% (28 out of 40). The two-sample t-test showed no significant differences between *L_1_* and *L_2_* (*p* = 0.786).

### 3.3. Time Comparison

The time consumed was recorded for both manual and AI processing methods. In each image, the average time for manual upper airway segmentation was 6 min 28 s. The time to average manual CSAmin determination was 21 min 10 s. With respect to AI processing, fully automatic algorithms of all four models can complete the segmentation and localisation tasks in the test group (all 40 images) within 1 s.

## 4. Discussion

With this study, we successfully explored the possibility that the AI models we developed were able to predict 3D information (CSAmin) from 2D data to solve clinical challenges. Regarding AI applications, airway segmentation and CSAmin localisation tasks are basic and initial steps in the development of AI systems for pathology detection and disease diagnosis in airway assessment. From the clinical perspective, this study provides an innovative upper airway analysis method that will help improve clinicians’ efficiency.

Nowadays, a few studies have explored and discussed the possibilities of applying AI techniques in upper airway analysis [26,27,28,29,30]. For example, Çağla et al. [31] developed an automatic pharyngeal airway segmentation in CBCT images by using a CNN system; a dice ratio of 0.919 and a weighted IoU of 0.993 were achieved. Antonio et al. [32] developed a fully automatic and end-to-end airway segmentation method based on the U-Net architecture, and the results showed the EXACT’09 public dataset achieved the highest sensitivity among three different test sets. And Rosalia et al. [33] applied a deep learning-based fully CNN technique to segment the sinonasal cavity and pharyngeal airway. A mean volume difference of 1.93 ± 0.73 cm^3^ was found between manual segmentation and AI processing.

However, most studies used 3D CT or CBCT images as training data to develop algorithms and AI models. In dentistry, only patients with severe craniomaxillofacial deformities are required to undergo CT or CBCT scans owing to the relatively higher cost and radiation dose than 2D radiography. Compared with 3D techniques, 2D images are more commonly applied in daily dental practice, which makes them an important tool for the early detection of potential airway problems. There have never been similar studies reported to infer CSAmin information from CBCT scans based on 2D data. Our study is the first to establish a fully automatic AI-driven system to evaluate the upper airway using 2D images. Compared with the detection of craniofacial hard tissues such as the skull, teeth and alveolar bone, airway structure detection is quite complicated owing to the complexity of the structures, uncertainty of landmarks and slight differences in the greyscale between air and soft tissue [34,35,36]. Given the sharing of convolution kernel parameters and sparsity of connections between layers, the CNN deep learning algorithms developed in this study allowed computer models comprising multiple convolutional layers to efficiently extract the boundaries and features of upper airway regions.

It is also extremely time-consuming to manually determine CSAmin, a crucial parameter in airway assessment, in three dimensions; the judgement cannot be based solely on 2D images because data from the third dimension are lacking. However, CNNs can predict 3D information from 2D data after realising a large amount of 3D feature data [24,25]. We utilised this strength of CNN deep learning algorithms and successfully developed a fully automatic AI-driven system for upper airway segmentation and CSAmin localisation using 2D radiographic airway images.

### 4.1. AI-Driven Segmentation Accuracy

The developed AI system was accurate and rapid in terms of upper airway segmentation. Regarding segmentation accuracy, all the four AI models achieved a precision of >90.0% and recall of >88.0% in the entire upper airway. Because of the lack of comparable studies on airway segmentation, we compared our accuracy with that reported in other studies on tooth detection involving deep learning methods and CNNs. Chen et al. [37] developed a fast region CNN method for detecting and numbering teeth in dental periapical films and reported a precision and recall of approximately 90%. Leite et al. [16] evaluated the performance of a new AI-driven system for tooth detection and segmentation and reported a sensitivity and precision of up to 90%. In addition to the high consistency with the above-mentioned studies, our segmentation models were ‘fully automatic’, whereas their AI segmentation systems required manual modifications after AI processing.

Owing to the irregular morphology and high variability in the nasopharynx region, the accuracy of segmentation in the nasopharynx was relatively lower than that in the retropalatal pharynx and retroglossal pharynx. Because the retropalatal and retroglossal segments are the main obstruction sites in the upper airway [38], the accuracy of segmentation in these two segments is critical for upper airway assessment. Our results showed that the precision of segmentation in the retropalatal pharynx and retroglossal pharynx was 92.0%, providing reasonably accurate anatomical information for clinical reference.

No differences were noted in the accuracies of the four AI segmentation models. The AI models with more convolutional layers (DeepLab101 and UNet36) did not show significantly better performance than the DeepLab50 and UNet18 models, perhaps because 2D data are relatively ‘simple’, with little heterogeneity and diversity. Therefore, the limited convolutional layers were sufficient for segmentation tasks and achieving good performance. For more comprehensive tasks, algorithms with a higher number of convolutional layers may show obvious superiority.

### 4.2. AI-Driven CSAmin Localisation Accuracy

Automatic CSAmin localisation was achieved using the established segmentation systems. In spite of the significance of CSAmin, the automatic localisation of CSAmin from 2D images is challenging because of a lack of 3D data, which renders manual determination impossible. In this study, we developed a new AI-driven algorithm to automatically predict the CSAmin height using 2D airway images, thereby facilitating the evaluation of CSAmin in LCs. The kappa test revealed a high consistency between manual processing using 3D images and AI prediction from 2D images (0.944). Our study therefore successfully validated the AI algorithms developed to accurately determine the CSAmin localisation based on 2D images.

Based on the results of our test set of 40 images, in 27 (67.5%) and 13 (32.5%) images, CSAmin localised to the retropalatal region and retroglossal region, respectively, consistent with previous retrospective clinical studies that demonstrated that the retropalatal region was the most common site for airway collapse in patients with OSA [39,40].

The calculated height errors of CSAmin localisation obtained from the 2D images were merely 1.95 and 2.53 mm for the manual and AI processing methods, respectively. The height error differences between the two segmentation ground truths demonstrated that the airway structure boundaries were slightly overestimated or underestimated during the AI processing. However, a height error of 2 mm was equivalent to 4–6 slices out of over 500 in a CBCT scan (0.8–1.2% of error), which did not affect the accuracy of CSAmin determination by the AI models in specific segments (kappa value: 0.944).

### 4.3. Limitations and Prospects

The purpose of using 3D CBCT scans in this study was to obtain exact CSAmin data. Considering ethical considerations in clinical practice that CBCT scans and LCs cannot be ethically obtained simultaneously from the same patient, we used 2D upper airway midsagittal images obtained from CBCT scans as the training dataset to establish CNNs. Previous studies have reported that both LCs and CBCT scans are reliable for the evaluation of upper airway structures and that there are no significant differences in the dimension measurements between the two techniques [23,41]. Although 2D midsagittal plane images are not an ideal source of training data, our study successfully established AI algorithms and systems with reasonably high accuracy. Further studies should consider the application of LCs to AI upper airway assessment to verify the practicality of the method.

In future, a larger number of subjects with different ethnicities, genders and ages should be recruited to develop more robust algorithms. In terms of future clinical applications, various learning material datasets, including those with clinical information, should be used to develop customised airway assessment systems for individual subjects.

## 5. Conclusions

In conclusion, we successfully developed an accurate and efficient fully automatic AI-driven system for upper airway segmentation and CSAmin localisation based on 2D radiographic upper airway images. The AI models we developed are able to predict 3D CSAmin information from 2D images for screening out potential airway problems, which provides an innovative upper airway analysis method that can improve clinicians’ efficiency greatly.

## Figures and Tables

**Figure 1 bioengineering-10-00915-f001:**
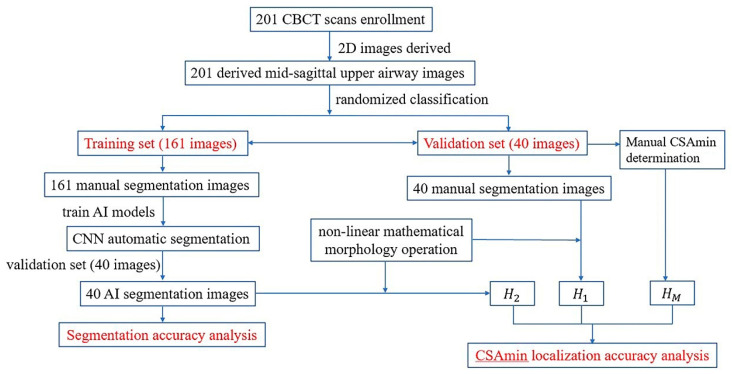
The dataset collection and flow of AI processing.

**Figure 2 bioengineering-10-00915-f002:**
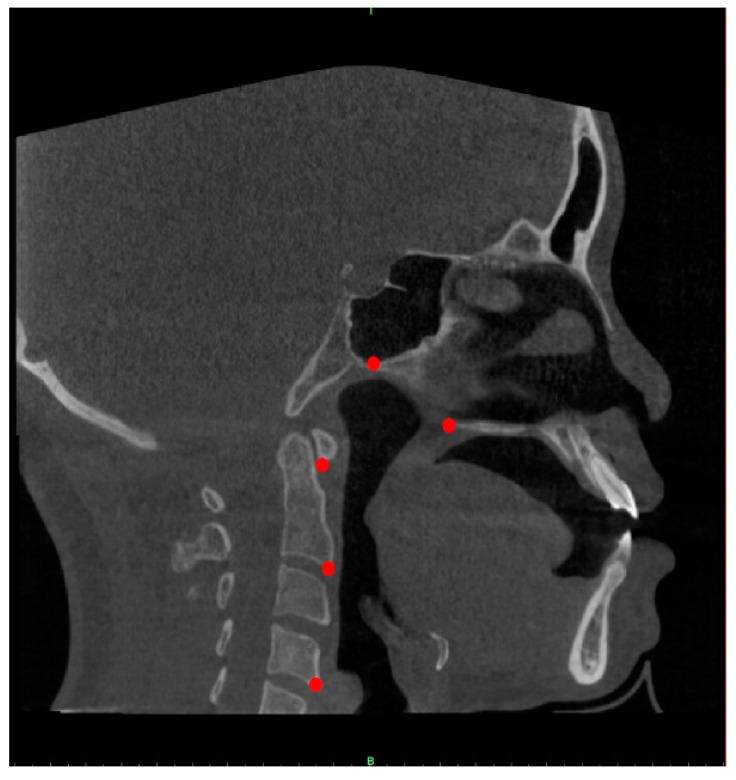
Landmark points on midsagittal plane images (PNS, VP, CV1, CV2 and CV4).

**Figure 3 bioengineering-10-00915-f003:**
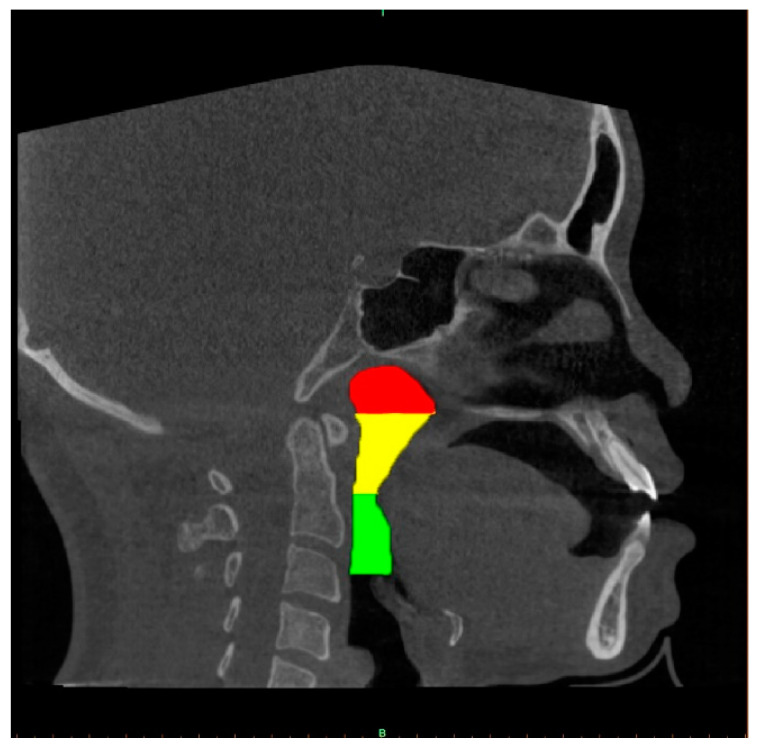
Three parts of the upper airway were annually marked in different colours. The red, yellow and green colour regions correspond to the nasopharynx, the retropalatal pharynx and the retroglossal pharynx, respectively.

**Figure 4 bioengineering-10-00915-f004:**
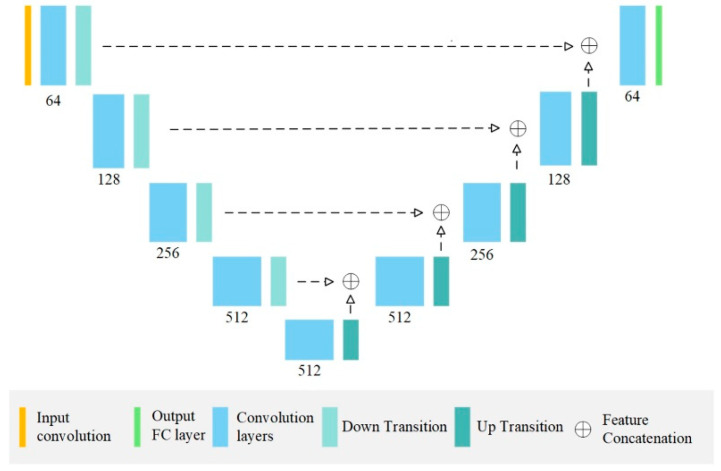
The architecture of the employed UNet model. Each convolution block contains 2 and 4 convolution layers (as well as batch-norm layers and ReLU activation units) for UNet18 and UNet36, respectively.

**Figure 5 bioengineering-10-00915-f005:**
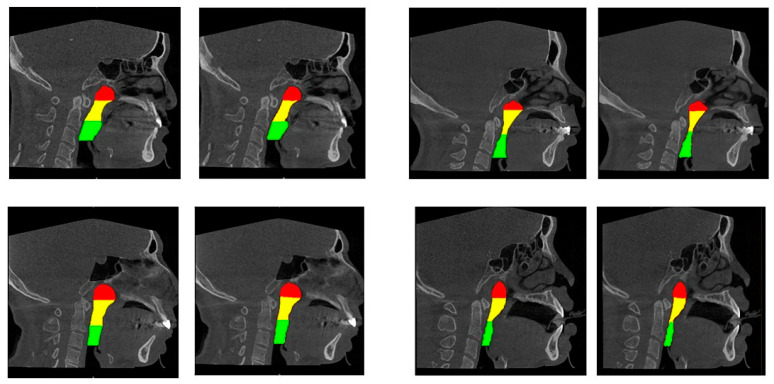
Manual segmentation results (**left**) and UNet18 model segmentation result (**right**). The red, yellow and green colour regions correspond to the nasopharynx, the retropalatal pharynx and the retroglossal pharynx, respectively.

**Figure 6 bioengineering-10-00915-f006:**

AI-driven results of the CSAmin localisation in GT1 (**left**) and GT2 groups (**right**).

**Figure 7 bioengineering-10-00915-f007:**
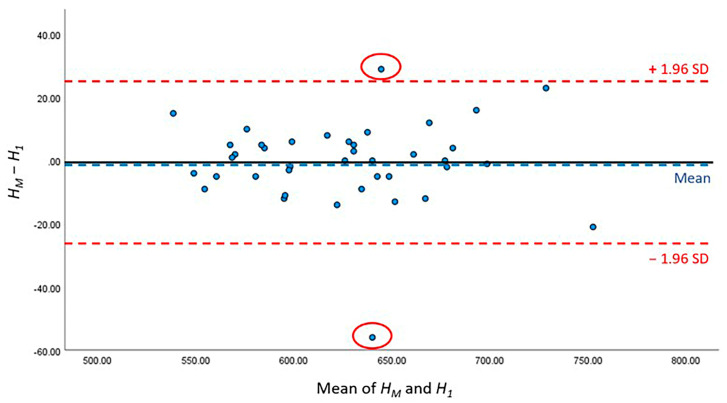
Bland–Altman figure of CSAmin height level between *H_M_* and *H_1_*.

**Figure 8 bioengineering-10-00915-f008:**
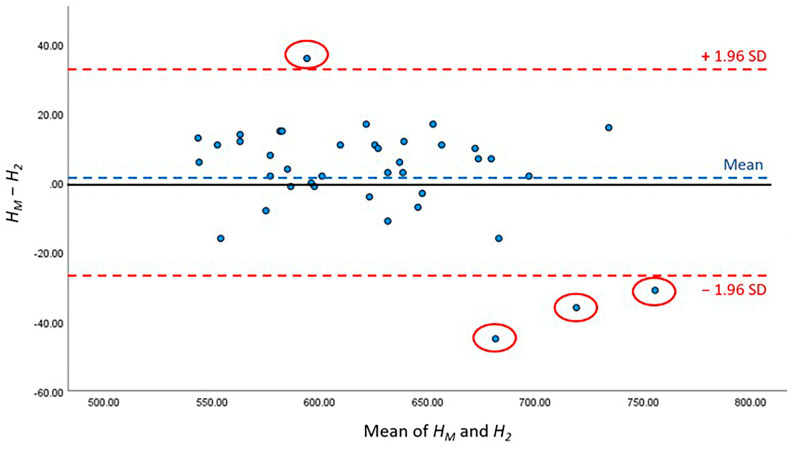
Bland–Altman figure of CSAmin height level between *H_M_* and *H_2_*.

**Table 1 bioengineering-10-00915-t001:** Definitions and abbreviations of upper airway landmarks in midsagittal plane.

Reference Points	Explanation
PNS	Most posterior point of palate
VP	Most posterior point of vomer
CV1	Most anterior inferior point of anterior arch of atlas
CV2	Most anterior inferior point of anterior arch of second vertebra
CV4	Most anterior inferior point of anterior arch of fourth vertebra

PNS = posterior nasal spine, VP = posterior point of vomer, CV1 = the first cervical vertebra, CV2 = the second cervical vertebra, CV4 = the fourth cervical vertebra.

**Table 2 bioengineering-10-00915-t002:** Performance (accuracy metrics) of the DeepLab system for segmenting the upper airway in 2D images. IoU = Intersection over Union, DSC = dice similarity coefficient.

Model	DeepLab50					DeepLab101				
	Precision	Recall	IoU	DSC	Size Difference	Precision	Recall	IoU	DSC	Size Difference
Nasopharynx	93.1	79.9	75.4	85.5	146.1	92.7	82.1	77.0	86.7	122.3
Retropalatal pharynx	87.3	90.4	79.8	88.6	102.1	89.1	89.5	80.6	89.1	101.1
Retroglossal pharynx	89.7	93.6	84.2	91.3	100.4	90.0	94.3	85.0	91.8	92.2
Overall	90.0	88.0	79.8	88.4	116.2	90.6	88.6	80.9	89.2	105.2
*p* value	0.369	0.381	0.415	0.488	0.039 *	0.451	0.467	0.434	0.451	0.322

**Table 3 bioengineering-10-00915-t003:** Performance (accuracy metrics) of the UNet system for segmenting the upper airway in 2D images. IoU = Intersection over Union, DSC = dice similarity coefficient.

Model	UNet18					UNet36				
	Precision	Recall	IoU	DSC	Size Difference	Precision	Recall	IoU	DSC	Size Difference
Nasopharynx	90.8	85.5	78.8	87.9	83.0	90.3	85.9	78.6	87.7	90.5
Retropalatal pharynx	87.6	90.8	80.4	88.9	115.0	88.8	89.8	80.5	89.0	102.8
Retroglossal pharynx	92.0	91.3	84.4	91.3	101.5	91.3	91.0	83.6	90.7	101.8
Overall	90.2	89.2	81.2	89.4	99.8	90.1	88.9	80.9	89.1	98.4
*p* value	0.469	0.398	0.416	0.433	0.270	0.416	0.488	0.498	0.433	0.807

**Table 4 bioengineering-10-00915-t004:** Consistency of CSAmin localisation in specific segments of upper airway between manual work and AI prediction in test group.

CSAmin Localisation	Manual Work		Total
	Retropalatal pharynx	Retroglossal pharynx	
AI prediction			
Retropalatal pharynx	27	0	27
Retroglossal pharynx	1	12	13
Total	28	12	40

**Table 5 bioengineering-10-00915-t005:** Performance of AI-driven CSAmin localisation results in GT_1_ and GT_2_ groups. SD, standard deviation.

Patient ID	*H_T_* (Pixel)	*H*_1_ (Pixel)	*H*_2_ (Pixel)	*L*_1_ (Pixel)	*L*_2_ (Pixel)	*L*_1_ (mm)	*L*_2_ (mm)
1	701	685	737	16	36	3.52	7.91
2	642	633	649	9	7	1.98	1.54
3	612	668	576	56	36	12.31	7.91
4	547	551	541	4	6	0.88	1.32
5	597	599	598	2	1	0.44	0.22
6	589	601	574	12	15	2.64	3.3
7	587	583	583	4	4	0.88	0.88
8	640	645	634	5	6	1.1	1.32
9	742	763	726	21	16	4.62	3.52
10	662	660	651	2	11	0.44	2.42
11	550	559	537	9	13	1.98	2.86
12	615	629	604	14	11	3.08	2.42
13	640	640	637	0	3	0	0.66
14	590	601	575	11	15	2.42	3.3
15	571	569	579	2	8	0.44	1.76
16	578	583	576	5	2	1.1	0.44
17	633	628	630	5	3	1.1	0.66
18	569	568	557	1	12	0.22	2.64
19	558	563	547	5	11	1.1	2.42
20	645	658	633	13	12	2.86	2.64
21	675	663	691	12	16	2.64	3.52
22	581	571	573	10	8	2.2	1.76
23	740	717	771	23	31	5.05	6.81
24	631	625	620	6	11	1.32	2.42
25	632	629	622	3	10	0.66	2.2
26	630	639	613	9	17	1.98	3.74
27	677	679	667	2	10	0.44	2.2
28	626	626	637	0	11	0	2.42
29	621	613	625	8	4	1.76	0.88
30	546	531	562	15	16	3.3	3.52
31	596	599	596	3	0	0.66	0
32	570	565	556	5	14	1.1	3.08
33	602	596	600	6	2	1.32	0.44
34	683	679	676	4	7	0.88	1.54
35	677	677	670	0	7	0	1.54
36	698	699	696	1	2	0.22	0.44
37	659	630	704	29	45	6.37	9.89
38	661	673	644	12	17	2.64	3.74
39	586	581	587	5	1	1.1	0.22
40	646	651	649	5	3	1.1	0.66
Mean	625.13	625.73	622.58	8.85	11.50	1.95	2.53
SD	50.24	50.38	56.27	10.07	10.08	2.21	2.21

## Data Availability

All data of the study are available from the Division of Paediatric Dentistry and Orthodontics, Faculty of Dentistry, the University of Hong Kong, and were used under license and approved by the Institutional Review Board of the University of Hong Kong/Hospital Authority Hong Kong West Cluster (IRB reference no: UW 21-519). The data are, however, available from the corresponding author upon reasonable request and with permission of the University of Hong Kong. Informed consent was obtained from all subjects involved in the study. Written informed consent has been obtained from the patient(s) to publish this paper.

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
