# Peer review of "Deep Learning Models for Automatic Upper Airway Segmentation and Minimum Cross-Sectional Area Localisation in Two-Dimensional Images"

_bioengineering, 2023, doi:10.3390/bioengineering10080915_

Round 1

Reviewer 1 Report

This study is about deep learning models for automatic upper airway segmentation by developing and validating CNN algorithms in 2D airway images. Developing and validating four different AI models for automatic upper airway segmentation is a complex task that requires expertise in deep learning, computer vision, and medical imaging.

The manuscript is well written. However, it can be further improved.

There are only few comments here:

Page 3, line 110. What is the ROI for the CBCT scan field view?

Page 4, line 154. How did the authors decide which slice of the mid-sagittal plane image to be used?

Page 5, line 189. All of the processing work was conducted in a single Nvidia RTX 2080 Ti GPU with 12G memory. The authors just mentioned the video card and RAM. Is it laptop or PC? How about the specification of the processor?

Page 7, line 266. Why Kruskal–Wallis test was chosen for the size differences? The data is not normally distributed?

Page 7, line 269. Why the paired-t test analysis was used to compare HM, H1 and HM, H2, instead of independent t-test?

Page 11, line 329. The average time to upper airway segmentation using AI processing methods was 6 min 28 s. Which AI method? All four models?

Moderate editing is required

Author Response

Comments 1: Page 3, line 110. What is the ROI for the CBCT scan field view?

Response: Thanks for your comment. The ROI of training data in the study are three segmented regions (the nasopharynx, retropalatal pharynx, and retroglossal pharynx) of upper airway in the annotated segmentation map. We stated this in Page 5, lines 172 – 176. And we added that point in Page 3, lines 108 – 110 in the revised manuscript to make this point more clear.

Comments 2: Page 4, line 154. How did the authors decide which slice of the mid-sagittal plane image to be used?

Response: Thanks for your comment. Five reference points (posterior nasal spine, posterior point of vomer and first, second and fourth cervical vertebra) were manually obtained by one examiner (CHU G) for each scan to determine the mid-sagittal planes of the CBCT images (Fig 2). We have made revision to state this point more clearly (please refer to Page 4, lines 147 – 150). Besides, the reliability and repeatability of the mid-sagittal plane selection was tested and the results are shown in Page 8, lines 274 – 275.

Comments 3: Page 5, line 189. “All of the processing work was conducted in a single Nvidia RTX 2080 Ti GPU with 12G memory”. The authors just mentioned the video card and RAM. Is it laptop or PC? How about the specification of the processor?

Response: Thanks for your comment. All the processing work was conducted on a PC with an Intel i7-8700 CPU, 32GB RAM and a single Nvidia RTX 2080 Ti GPU with 11G VRAM. We revised the manuscript by stating this point more clearly (please refer to Page 5, lines 185 – 187).

Comments 4: Page 7, line 266. Why Kruskal–Wallis test was chosen for the size differences? The data is not normally distributed?

Response: Thanks for your comment. As the size difference results of one group (UNet 36) was not normally distributed after testing by Shapiro-Wilk test (p = 0.037), we used Kruskal–Wallis test for the size differences comparison. And we have added this point in 3.1 Results part (Page 8, lines 284 – 286) in the revised manuscript.

Comments 5: Page 7, line 269. Why the paired-t test analysis was used to compare HM, H1 and HM, H2, instead of independent t-test?

Response: Thanks for your comment. The comparisons of HM, H1 and HM, H2 were calculated or predicted height results derived from the same test group (40 images). Therefore, we used paired-t test analysis as they shared the same sample.

Comments 6: Page 11, line 329. The average time to upper airway segmentation using AI processing methods was 6 min 28 s. Which AI method? All four models?

Response: Thanks for your comment and figuring out the typo. The average time for manual upper airway segmentation in each image was 6 min 28 s. While in all four AI models, the fully automatic algorithms can complete the segmentation and localisation tasks in the test group (all 40 images) within 1 s. We have corrected the sentence in Page 11, lines 329 – 333.

Reviewer 2 Report

The topic is very interested , but there are major issues in the paper

the organization of the paper is miss leading, and the figures are blurred, no explanation well about the method , result and the conclusion . No comparison   with the LR . The paper is not on MDPI formats .

It needs more and more work then resubmit it to review it again

Author Response

Comment 1: the organization of the paper is miss leading, and the figures are blurred, no explanation well about the method , result and the conclusion . No comparison with the LR . The paper is not on MDPI formats .

Response: Thanks for your comments.

We have revised the manuscript using MDPI formats. And we have revised the resolution of all figures at least 330 dpi.

We have revised the structure of manuscript based on abstract, introduction, materials and methods, results, discussion, conclusions and references. Besides, we stated the research gap and clinical novelty more clearly in Page 2, lines 86 – 91 and Page 11, lines 335 – 340, respectively. Moreover, we used subheading to make the organization of the manuscript clearer.

To make the structure clearer, in materials and methods, we used seven subheadings to illustrate data collection, consistency tests, manual determination of the mid-sagittal plane and image capture, two main tasks (upper airway segmentation and CSAmin localisation), time comparison and statistical analysis. In results part, we also used three parts to illustrate the main results of the research (1. upper airway segmentation results, 2. CSAmin localisation results, and 3. time comparison results). In discussion part, we added three subheadings 4.1 AI-driven segmentation accuracy (Page 12, line 357), 4.2 AI-driven CSAmin localisation accuracy (Page 12, line 383) and 4.3 Limitations and prospects (Page 13, line 404) to make the discussion part clearer.

As for “no comparison with the LR”, considering ethical considerations in clinical practice that CBCT scans and LCs cannot be ethically obtained simultaneously from the same patient, we used 2D upper airway mid-sagittal images obtained from CBCT scans as the training dataset to establish CNNs. And one of the aims in the study was we tried to explore the possibility of whether the developed AI models can help to segment upper airway and predict 3D information (CSAmin) from 2D data, even though 2D mid-sagittal plane images are not an ideal source of training data, our study successfully established AI algorithms and systems with reasonably high accuracy, we hope the models can be applied to assist clinicians in daily clinical practice. We have added this part of discussion in the revised manuscript (Page 13, line 406 – 413).

Reviewer 3 Report

The research validated convolutional neural network algorithms for automatic upper airway segmentation and minimum cross-sectional area (CSAmin) localisation using 2D radiographic airway images. Four AI models were trained, and their performance was evaluated on a test group and a validation group. The precision of all four models exceeded 90%, with no significant differences in accuracy between them. The fully automatic AI-driven system was significantly more efficient than manual processing in airway segmentation and CSAmin localisation.

My main concerns are as follows:
- Literature review: The authors claim that "the effects of AI techniques on upper airway assessment have not been thoroughly explored". However, I noticed that there are vast and remarkable research studies on similar topic, including:
    - Garcia-Uceda, A., Selvan, R., Saghir, Z. et al. Automatic airway segmentation from computed tomography using robust and efficient 3-D convolutional neural networks. Sci Rep 11, 16001 (2021).
    - Maken, Payal, Abhishek Gupta, and Manoj Kumar Gupta. A systematic review of the techniques for automatic segmentation of the human upper airway using volumetric images. Medical & Biological Engineering & Computing (2023): 1-27.
    - Sin, ÇaÄŸla, et al. A deep learning algorithm proposal to automatic pharyngeal airway detection and segmentation on CBCT images."Orthodontics & Craniofacial Research 24 (2021): 117-123.
- Contribution. Most of this research focuses on barely comparing four neural networks. This is apparently not sufficient for academic publication.
- Presentation. Inconsistent font size in the text, especially line 141-147.

Author Response

Comment 1: The authors claim that "the effects of AI techniques on upper airway assessment have not been thoroughly explored". However, I noticed that there are vast and remarkable research studies on similar topic.

Comment 2: Contribution. Most of this research focuses on barely comparing four neural networks. This is apparently not sufficient for academic publication. 

Response: Thanks for your comments 1 and 2 which are related to the research gap and novelty of this study.

Although the technology of AI and CNNs are mature and have been investigated in image analysis, the research gap is there were no similar studies reported to successfully infer CSAmin information based on 2D data. Therefore, our study aimed to use the combination of image segmentation processing and mathematical morphological operations for automatic upper airway segmentation and CSAmin localisation. And the most novelty of the present study is we explored the possibility of whether the developed AI models can help to segment upper airway and predict 3D information (CSAmin) from 2D data to solve clinical challenges since many patients do not have 3D image but 2D image as the initial examination. Therefore, we hope the models can be applied to assist clinicians in daily clinical practice by “screening” out the cases which need further attention. In the revised manuscript, we have added the clarification of the research gap and clinical novelty in Page 2, lines 86 – 91 and Page 11, lines 335 – 340, respectively.

Besides, the results of this study  consist three main parts. First, we achieved automatic upper airway segmentation based on CNNs and compare the performance among four neural networks. Second, we achieved automatic CSAmin localisation by mathematical morphological operations. And finally, we compared the efficiency in two tasks between manual work and AI processing. Therefore, our results would be beneficial for AI application scenarios as well as for clinical dentistry.

Comment 3: Presentation. Inconsistent font size in the text, especially line 141-147.

Response: Thanks for your comments. We have checked and revised font size in the whole manuscript to make the font size and format to be clear and consistent.

Round 2

Reviewer 2 Report

The paper needs more expanding in conclusion part 

Author Response

Comments 1: The paper needs more expanding in conclusion part.

Response: Thanks for your comment. We revised the conclusion part (Page 13, lines 437 – 442) to make the conclusions more comprehensive and clearer.

Reviewer 3 Report

Thanks for the authors' response.

As I mentioned in my last review report, there are "vast and remarkable research studies on similar topic". That means, as far as I know, there are several outputs that also focus on digging out using machine learning methods to segment or localize upper airway structures using similar data. I strongly recommend the authors search for and quote them to discuss why the proposed method potentially outperforms them. Unfortunately, the authors didn’t mention them. Consequently, I cannot find out the contribution of this research.

Author Response

Comment 1: As I mentioned in my last review report, there are "vast and remarkable research studies on similar topic". That means, as far as I know, there are several outputs that also focus on digging out using machine learning methods to segment or localize upper airway structures using similar data. I strongly recommend the authors search for and quote them to discuss why the proposed method potentially outperforms them. Unfortunately, the authors didn’t mention them. Consequently, I cannot find out the contribution of this research.

Response: Thanks for your comments. We added a new paragraph in discussion part (Page 12, lines 342 – 358) to discuss the difference and novelty of our research when compared to previous studies. And we cited the related publications in references part (reference 26 – 33).